# Developing and Testing the Air Cooling System of a Combined Climate Control Unit Used in Pig Farming

Ivan Ignatkin [1,*], Sergey Kazantsev [1], Nikolay Shevkun [1], Dmitry Skorokhodov [1], Nikita Serov [1], Aleksei Alipichev [1] and Vladimir Panchenko [2]

1   Moscow Timiryazev Agricultural Academy, Russian State Agrarian University, Moscow 127550, Russia
2   Department of Theoretical and Applied Mechanics, Russian University of Transport, Moscow 127994, Russia
*   Correspondence: ignatkin@rgau-msha.ru

**Abstract:** This article presents the results of developing and testing the air-cooling system of a combined climate control unit used in pig farming. The authors have found a water-evaporative system to be the most efficient for cooling the air supply. Cooling systems of this type consume 0.003 kW/kW of electric power to produce 1 kW of cold. Based on the developed mathematical model for water-evaporative cooling in the combined climate control unit, the authors have determined that an air supply with a temperature of 31.2 °C and a relative humidity of 30.4% can be cooled by 8.3 °C when saturated with moisture to a relative humidity of 90.0% (by 11.7 °C at 100%). Experimental studies of the cooling system confirmed the theoretically obtained data.

**Keywords:** indoor climate; air cooling; water-evaporative systems; sprayed panels; heat recovery units

## 1. Introduction

Pig breeding is a highly productive branch of animal husbandry due to fast-growing pig breeds. Thanks to its nutritional value, availability and extraordinary cooking characteristics, pork meat plays a significant role in the human diet. In Russia, pork meat accounts for 25.6% of the meat resources consumed. The number of pigs in the Russian Federation amounted to 25.9 million by the end of 2021, 90.2% of which were industrially grown [1]. Such enterprises use flow technologies, high animal concentration, and intensive use of breeding stock to achieve maximum productivity. All these factors require a high level of production at all stages. Indeed, any deviation from the optimum production mode inevitably leads to losses.

The indoor climate is determined by a combination of temperature, relative humidity, chemicals, air composition, contaminants, micro-organisms, light, etc. Each of these indicators has a significant impact on animal productivity and must be maintained within strict limits based on the physiological needs of the animals. The most important indicators are temperature and relative humidity [2]. It is reasonable to use these indicators as regulators for the heating and ventilation system.

The body of a pig is covered with very sparse wool. It does not actually protect against external temperature influences. A stable body temperature is maintained by the thermoregulation system, in which the body uses energy to maintain a constant temperature. This energy consumption rate is minimal at an optimum temperature (Figure 1) [3].

Currently, genetics companies have significantly lowered the fat content of pork meat by reducing the thickness of the subcutaneous tissue that acts as natural thermal insulation in pigs. As a result, breeds developed with modern genetic techniques are more sensitive to temperature drops.

Figure 2 shows that a high relative humidity ($\varphi > 75\%$) results in a decrease in pig weight gain (by 20%) and an increase in feed consumption (by 40%) [3].

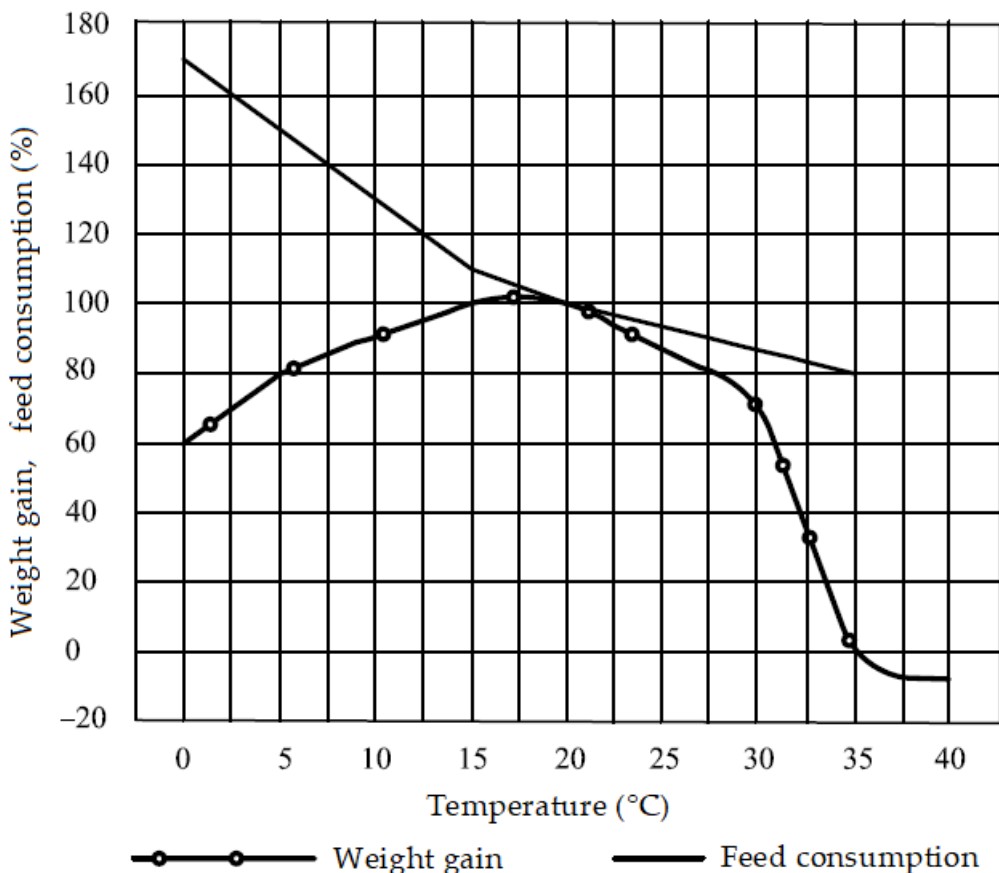

**Figure 1.** Influence of ambient temperature on the performance of fattened pigs.

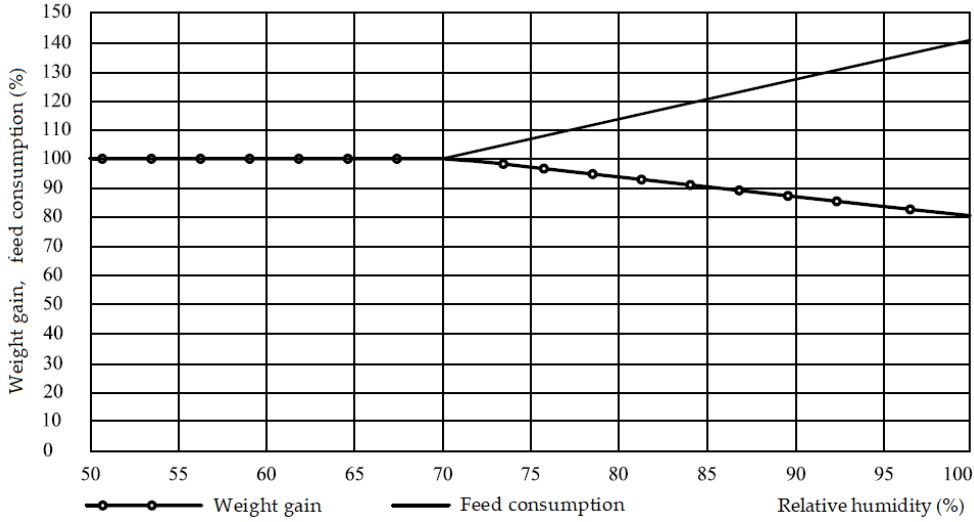

**Figure 2.** Effect of relative humidity on Large White pig productivity.

Dry air (relative humidity under 50%) also negatively affects the animal's body, causing irritation of the mucous membranes of the eyes and the respiratory tract, decreased local immunity, increased thirst, and consequently reduced appetite and nutrient absorption.

The above materials show the considerable influence of indoor climate on the productivity of animals.

Indoor climate indicators and animal excretion rates are taken by planners as input data from their manuals. It should be noted that the updated norms currently operating in

Russia—"RD-APK 1.10.02.04-12 methodological guidelines on engineering design of pig farms and facilities"—are based on the All-Union Science and Technical Regulations 2–96. The requirements have not changed much as to the emission of heat and harmful gases by animals and indoor climate parameters.

Russian pig breeders have achieved tremendous results in ensuring early maturity and increasing prolificacy [4]. The animal's body has become longer. The ratio of the surface area of a pig to its volume has changed; the thickness of the rump has decreased, which inevitably influences the parameters of its heat exchange with the environment. By comparing domestic and foreign excretion rates and indoor climate requirements, we can see a number of differences [5].

The entry "Nursing sows with piglets" in RD-APK provides data for animals weighing up to 200 kg, although the weight of a typical sow is 200–300 kg. These data are also available in the DIN 18910 standard operating in Germany (Berechnungs- und Planungs-grundlagen fuer das klima in geschlossenen ställen—norms for calculating parameters and indoor climate planning in closed stables (barns)).

The standard indoor climate parameters are also different, as shown in Table 1.

**Table 1.** Standard indoor climate parameters (data taken from RD APK and DIN 18910).

| Age and Gender Group | Weight, kg | Standard Parameters | | | |
|---|---|---|---|---|---|
| | | RD-APK 1.10.02.04-12 | | DIN 18910 | |
| | | Indoor Temperature, °C | Indoor Relative Humidity, % | Indoor Temperature, °C | Indoor Relative Humidity, % |
| Reproductive, barren, gestation sows, boars | Over 50 | 16 | 80 | 16–20 | 75 |
| Nursing sows with local heating for piglets | Over 100 | 18 | 80 | 20 | 75 |
| Pigs fattened using the "all empty—all occupied" method | 10 | 24 | 70 | 24 | 70 |
| | 20–30 | 20 | 80 | - | - |
| | 40–50 | 18 | 80 | - | - |
| | 60–100 | 16 | 80 | 18 | 70 |

The data presented in Table 1 show that the requirements are basically the same, but DIN 18910 is stricter with regard to the requirements for relative humidity and temperature. However, the $CO_2$ concentration requirements are stricter in the RD-APK and amount to 0.2%, compared with 0.3% in DIN 18910. In our opinion, this is due to the fact that in the Soviet Union central heating systems were used everywhere to provide the required concentration of carbon dioxide. However, decentralized heating systems operating on natural gas are more common now due to their lower cost, as described in the thesis written by D. Tikhomirov [6].

Open combustion heat generators direct the flue gases indoors. The main component of natural gas is methane. The products of its combustion include carbon dioxide and water. With these systems, a carbon dioxide concentration of 0.2% is theoretically unachievable in pig houses during the coldest period, but 0.3% is still achievable.

Climate control systems used in crop production [7] and livestock farming aim to achieve normative values for temperature, relative humidity, and pollutant concentration [8]. Temperature is an important indicator of the indoor climate in livestock buildings [9].

In winter, a good supply of heat is required for heating the air supply [10]. Therefore, exhaust air heat recovery systems offer an effective way to reduce heating energy consumption [11,12]. Supply and exhaust air units with heat recovery exchangers utilize the heat in the exhaust air without mixing exhaust and supply air [13], thus supplying

clean heated air to the production facilities. There are also ventilation systems with spiral recuperators [14,15] for heat recovery in winter and air cooling in summer.

Regenerative heat exchangers are widely used in the industry, e.g., in transcritical $CO_2$ heat pumps [16] and in $CO_2$ heat pump systems with compression/ejection for simultaneous cooling and heating [17].

In decentralized pit-type systems it is important to ensure reliable separation of supply and extract air [18].

However, in summer the challenge is no less important for decreasing the indoor temperature [19]. This is often achieved with individual cooling systems [20].

The importance of energy-saving systems using heat recovery can hardly be overestimated; they can be found in both mobile and stationary equipment [21]. In our opinion, it is reasonable to consider retrofitting supply and exhaust heat recovery units with cooling systems [22]. Such a solution would increase the intensity of equipment use, reducing the payback period.

However, in some cases the use of modular coolers is feasible [23].

This work aims to develop and test an air cooling system in a combined climate control unit used in pig farming.

To achieve this goal, it is necessary to solve the following tasks:

1. To analyze and classify the available air cooling ventilation systems;
2. To justify the construction of the combined climate control unit with the air cooling system;
3. To design a mathematical model of cooling in the climate control unit;
4. To experimentally test the developed system.

Air temperature can be decreased to the required values in different ways.

For pig farms, air cooling methods can be divided into two types: water-evaporative cooling systems and vapor-compression refrigerating units (Figure 3).

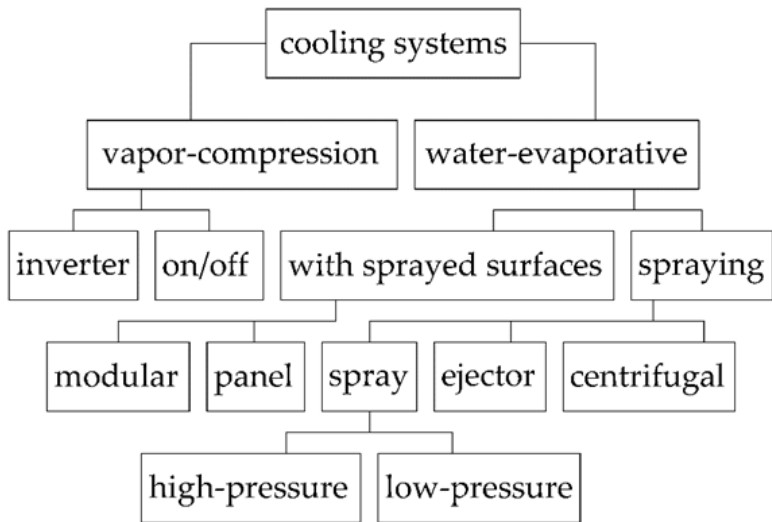

**Figure 3.** Classification of cooling systems.

The vapor-compression refrigeration unit is a closed hermetic system consisting of an evaporator, a compressor, a heat exchanger condenser, a filter-dryer and a throttle connected by pipelines. The unit is filled with a refrigerant with a boiling point lower than that of the cooled medium. The cooled air comes into contact with the surface of the evaporator and transfers its heat to the refrigerant, turning it into vapor. The compressor forces the refrigerant vapor into the condenser, where the latter is turned into its liquid state. Vapor-compression refrigeration systems are used extensively in industry and civil engineering, but they are rarely used in animal husbandry. However, where strict temperature limits are

set, regardless of weather conditions, vapor-compression refrigeration units are the only feasible solution.

With water evaporation cooling, the air is cooled by being blown through a spray chamber or a cartridge package soaked in water and subjected to consecutive heat exchanges. Another option is spraying water into the cooled air stream via nozzles. The evaporated moisture absorbs the heat of vapor formation and the air is cooled.

Water-evaporative cooling is fundamentally simpler and less demanding in terms of maintenance and operating conditions, but it has a number of limitations and its efficiency depends significantly on the temperature and relative humidity of the outdoor air.

In spite of their fundamental similarity, water-evaporative cooling systems are available in a wide range of designs. The most noteworthy are water spray systems and systems with sprayed surfaces. Table 2 presents the results of their comparative assessment.

**Table 2.** Analysis of cooling system efficiency.

| Parameter | Vapor-Compression Refrigerating Units (VCRU) | | Water-Evaporative Cooling Systems | | | | | |
|---|---|---|---|---|---|---|---|---|
| | VCRU On/Off | Inverter VCRU | Low-Pressure Nozzles (2 atm) | High-Pressure Nozzles (70 atm) | Sprayed Panel | Centrifugal Cooling Unit | Sprayed Modules | Ejector Modules |
| Supply air temperature reduction efficiency, °C | - | - | 1 | 3 | 15 | 6 | 8 | 4.5 |
| Cooling capacity of the system, kW | 282 | 282 | 34.2 | 102.6 | 424.9 | 170.0 | 226.6 | 127.5 |
| Installed electrical capacity of the cooling units, kW | 95.9 | 95.9 | 1 | 2.2 | 1.6 | 33.4 | 25.2 | 66.0 |
| Hourly energy consumption of the cooling units, taking into account irregular use, kWh | 95.9 | 67.1 | 0.8 | 2.2 | 1.2 | 33.4 | 25.2 | 66.0 |
| Set air exchange rate, m$^3$/h | 151,800 | 151,800 | 442,750 | 442,750 | 366,850 | 366,850 | 366,850 | 366,850 |
| Installed fan power, kW | 6.6 | 6.6 | 18.9 | 18.9 | 15.6 | 15.6 | 15.6 | 15.6 |
| Installed electric power of the system, kW | 102.4 | 102.4 | 19.9 | 21.1 | 17.2 | 48.9 | 40.8 | 81.6 |
| Electrical power consumption for producing 1 kW of cold, kW/kW | 0.340 | 0.238 | 0.023 | 0.021 | 0.003 | 0.196 | 0.111 | 0.518 |
| Refrigeration production efficiency criterion, kW/kW | 2.9 | 2.9 | 34.2 | 46.6 | 265.6 | 5.1 | 9.,0 | 1.9 |
| Water consumption, m$^3$/h | 0 | 0s | 44.2 | 0.7 | 3.7 | 2.5 | 3.7 | 0.2 |
| Specific energy consumption per 1000 m$^3$/h air, W | 631.6 | 442.1 | 1.8 | 5.0 | 3.3 | 90.9 | 68.8 | 179.9 |

Water is evaporated by absorbing the heat of vapor formation. The energy is thus spent to provide the required air exchange and water supply to the evaporation zone [24]. Ejection and nozzle systems do not directly consume energy for the cooling process. However, their operation requires water supplied at a given flow rate and pressure, which results in electricity costs at the water plant [25]. These costs have been taken into account in estimating the costs of electric power for the production of 1 kW of cold.

The following conditions and assumptions are taken into account when comparing cooling systems:

- vapor-compression cooling systems have unlimited cooling capacity under the operating conditions;
- the estimated air exchange of the water-evaporative cooling systems takes into account equal air exchange for the removal of excess heat and moisture.

Among the water-evaporative cooling systems, systems with sprayed surfaces are the most effective in terms of reducing the air supply temperature. They are also the most energy efficient.

Based on the above and considering the fact that the climate control unit has a large heat exchange surface area, it is advisable to spray water on it to use the effect of water-evaporative cooling of the supply air.

Water-evaporative cooling is widely used in the climate control systems of various production facilities due to its combination of low cost and high efficiency [26]. However, the system's operation is highly dependent on outdoor air parameters [24]. When designing climate control systems, it is important to analyze the main process indicators [27] and to predict the efficiency of water-evaporative cooling in different climate zones. In this connection, it is advisable to design a computational model of the combined unit [28] to link geometric parameters of the cooling unit based on the recuperative heat exchanger, its performance, total aerodynamic resistance of the system and air parameters at the inlet and outlet of the refrigerating unit.

## 2. Materials and Methods

### 2.1. Description of the System

The heat-exchanging surfaces are to be cleaned periodically in the process of operation. Consequently, the spraying system performs cleaning and disinfecting functions in addition to cooling [29]. This requires the use of detergent and/or disinfectant solutions. Disinfection can affect both the working surfaces of the unit and the ventilation air, making it possible to disinfect the supply and exhaust air of the serviced area.

The spraying system consists of a pipeline with nozzles, a time switch and a solenoid valve, and is connected to a water supply system for periodic water spraying in the exhaust and supply ducts above the heat exchanger. The sprayed liquid can be used in a flow-through or a cyclic mode. In the latter case, it requires a storage tank, which can take the form of a sump unit [30].

The offered solution ensures a more intensive use of the equipment, helps save on the number of cooling units and provides for the implementation of previously unavailable functions that are extremely important in terms of longevity, operational efficiency and biological safety—thus increasing the economic efficiency of the system as a whole.

The block diagram is shown in Figure 4.

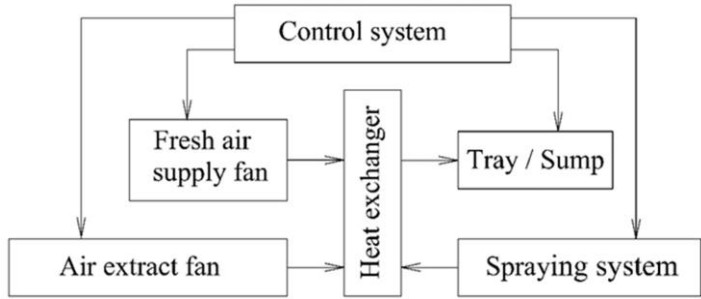

**Figure 4.** Schematic diagram of the combined climate control unit.

During the warm season, the ventilation system is used to remove excess heat, and the ventilation rate is significantly higher than in winter. In particular, in winter the specific air consumption per 1 kg of live weight of fattened piglets is 0.17 to 0.21 m$^3$/kg/h, and in summertime this value increases to 1.1 to 2.5 m$^3$/kg/h, that is, 5.2 to 14.7 times higher.

In warm periods, air cooling occurs as follows. The indoor air is extracted by separate window or roof fans. The exhaust fan of the unit is reversed, and both channels provide the air inflow. A process diagram of the prototype of the combined climate control unit equipped with the water-evaporative cooling system is shown in Figure 5. The surface of the heat exchanger is sprayed with water via nozzles. On contacting the heat exchanger

surface, the water wets it in a film or droplet pattern, creating a large cooling area. The air supply passes through the heat exchanger, comes into contact with the heat exchanger surface, is cooled there and then discharged into the room by the fans.

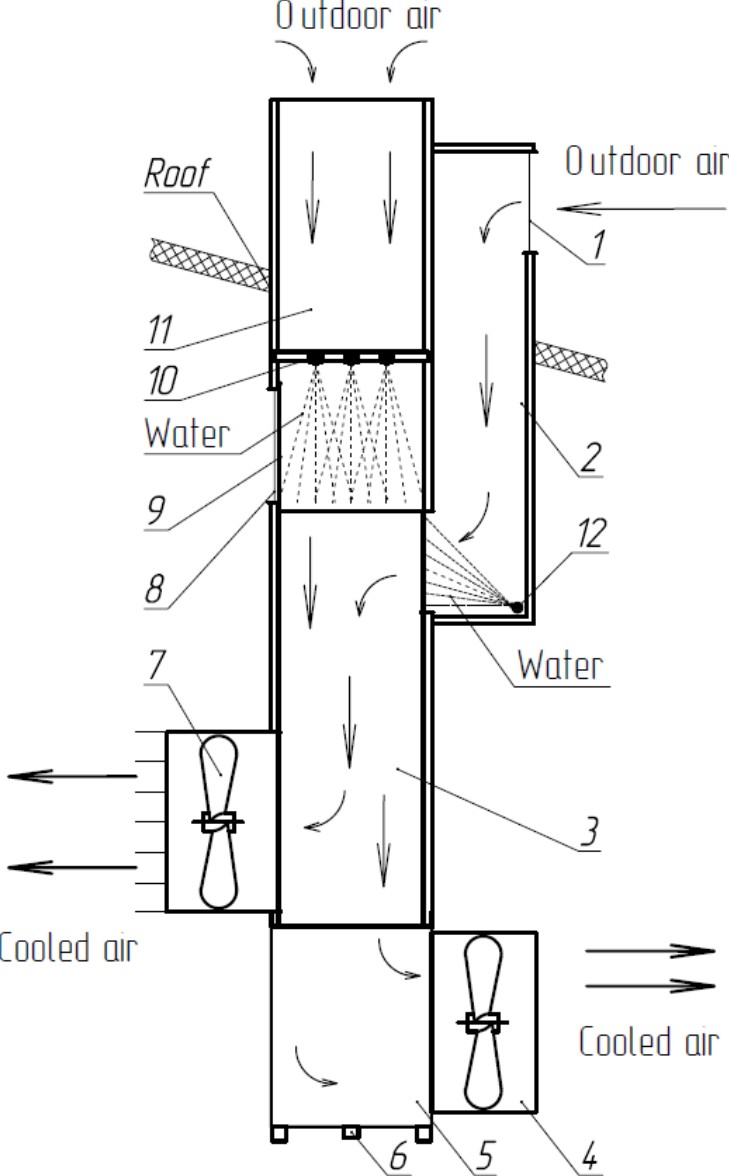

**Figure 5.** The cooling mode of the combined climate control system: 1—inlet window; 2—inlet duct; 3—heat exchanger; 4—exhaust fan; 5—sump; 6—discharge pipe; 7—supply fan; 8—recirculation opening; 9—recirculation damper; 10—pipeline with nozzles; 11—exhaust duct; 12—pipeline with nozzles.

*2.2. Theoretical Research*

Modelling of Water-Evaporative Cooling in the Unit

The water-evaporative cooling process in systems with sprayed layers can be represented as a convective heat exchange of the air flow with heat exchanger plates and a simultaneous evaporation of water from the unit surface accompanied by the absorption of vaporization heat.

The finite difference method is widely used to find a numerical solution to heat exchange problems. In a steady-state mode, the system is described by a set of parameters: the surface temperature of the unit (equal to the wet-bulb temperature), temperature

and relative humidity of the supply air. The air supply cools as heat and moisture are exchanged, approaching the cooling surface temperature, while its enthalpy (heat content) remains unchanged.

The process is limited by the exposure (the process duration), the relative humidity at the unit outlet (climate requirements) and the outdoor air parameters (the temperature and moisture content), which will determine the physical cooling potential.

For practical calculations, these parameters can be considered conditionally stationary. In the boundary layer, at the cooling surface, the relative humidity is 100%. The temperature of the unit wall is equal to the wet-bulb temperature and is assumed as constant for the considered conditions (Figure 6) [16].

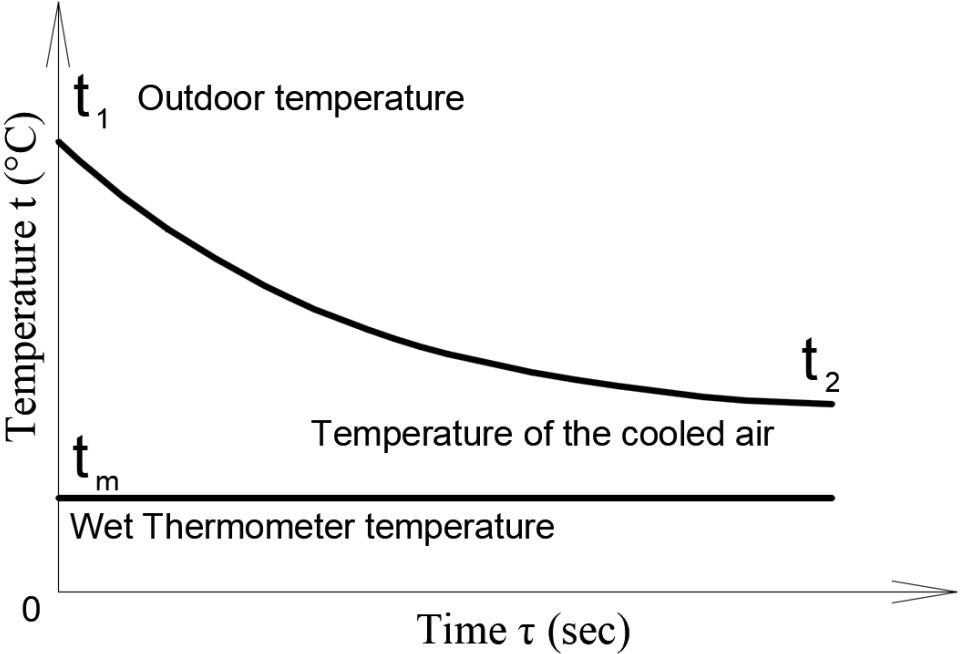

**Figure 6.** Diagram of water-evaporative air cooling.

The heat and mass transfer process is characterized by equality of heat extracted from the supplied air and the heat of vaporization expended for moisture evaporation from the unit surface [31]. The cooling intensity (heat transfer) of the air supply is determined based on Newton's equation

$$q_{\text{cool}} = S \cdot \overline{\alpha} \cdot (t_o - t_w), \tag{1}$$

where $S$—the heat exchange surface area, m²; $\overline{\alpha}$—the heat transfer coefficient average over the heat exchange surface, W/(m²·K); $t_o$—the outdoor air temperature, °C and $t_w$—the heat exchange surface (wet-bulb) temperature, °C.

The heat transfer coefficient $\alpha$ is the amount of heat transferred through the unit heat transfer surface per unit time at a temperature difference of 1 °C [32].

In engineering calculations, it is expedient to use criterion analysis to determine its value. Thus, based on the aerodynamic similarity criteria it is possible to determine the heat transfer coefficient with high accuracy [33].

$$q_{\text{cool}} = S \cdot \overline{\alpha} = Nu \frac{(t_o \cdot 7 \cdot 10^{-5} + 0.0237)}{d_{e.p}}, \tag{2}$$

where $Nu$—Nusselt criterion characterizing the correlation of the intensity of heat transfer by convection and heat conduction and $d_{e.p}$—hydraulic diameter of the pipe, m.

In the considered unit, the air ducts are pipes with a rectangular cross-section. The channels of the plates are directed along the main air flow (taken conditionally as vertical), as shown in Figure 7.

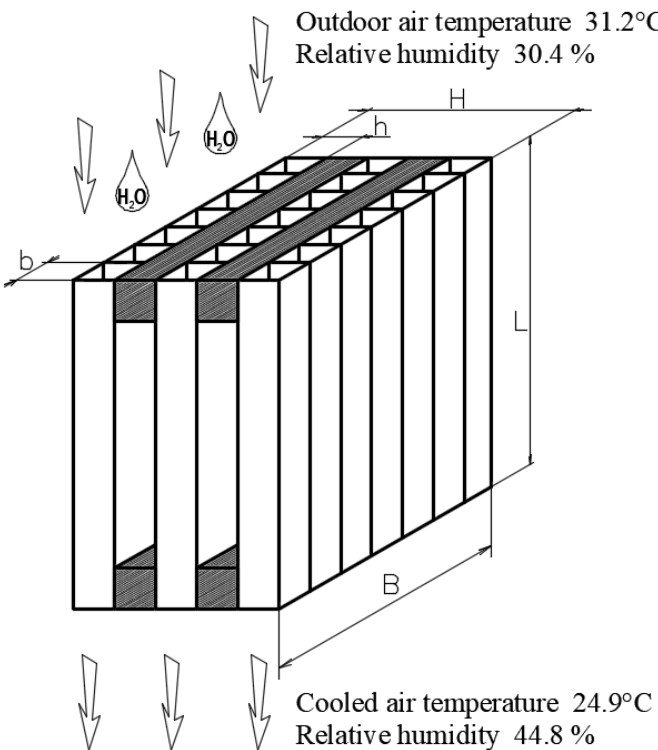

**Figure 7.** Heat exchanger in cooling mode.

The equivalent diameter for a pipe of rectangular cross-section is determined by the following formula ([32], p. 189):

$$d_{e.p} = 4\frac{A}{P} = 4\frac{b \cdot h}{2 \cdot (b+h)},\tag{3}$$

where $A = b \cdot h$—cross-section area, m$^2$; $P = 2 \cdot (b+h)$—perimeter, m; $b$—width of the tube base of a triangular cross-section, m and $h$—height of the tube of a triangular cross-section, m.

The Nusselt criterion ($Nu = f\left(\text{Pr}; \text{Re}; \frac{\Delta T}{T}\right)$) is defined by the following formula:

$$Nu = 0.28 \cdot \text{Re}^{0.6} \cdot \overline{\text{Pr}}^{0.36} \cdot \left(\frac{\overline{\text{Pr}}}{\text{Pr}_l}\right)^{0.25},\tag{4}$$

where Re—the Reynolds criterion value for the air flow in the cooling unit; $\overline{\text{Pr}}$—the average Prandtl criterion value for the air flow in the cooling unit and $\text{Pr}_c$—the Prandtl criterion value for the boundary layer.

The Reynolds criterion value is defined by the formula

$$\text{Re} = \frac{v \cdot d_{e.p}}{\nu_{su}},\tag{5}$$

where $\nu_{su}$—the average kinematic viscosity of the supply air, determined at an average air temperature in the unit $t_{av} = \frac{t_o + t_w}{2}$.

The Prandtl criterion value for the boundary layer is defined by the following formula:

$$\mathrm{Pr_l} = \frac{\eta_l \cdot c_m \cdot 10^3}{t_w \cdot 7 \cdot 10^{-5} + 0.0237}, \tag{6}$$

where $\eta_l$—dynamic viscosity of the layer, Pa·sec and $c_m$—specific mass heat capacity of the air, J/(kg·K).

The dynamic viscosity of the air in the boundary layer is determined by the following formula:

$$\eta_l = \nu_l \cdot \rho_{su}, \tag{7}$$

where $\nu c$—kinematic viscosity of air in the boundary layer, m$^2$/s and $\rho_{su}$—air density in the boundary layer, kg/m$^3$.

The temperature of the air in the boundary layer is asymptotically close to the wall temperature and numerically may be assumed to be equal to the wet-bulb temperature. In this case, the kinematic viscosity of air in the boundary layer can be found using the following formula [34]:

$$\nu_l = \frac{324 - 1.5 \cdot t_w}{10^9} p_0 + 16.81 + 0.048 \cdot t_w \cdot 10^{-6} \cdot 287 \frac{t_w + 273}{p_0}, \tag{8}$$

where $t_w$—wet-bulb temperature, °C and $p_0$—atmospheric pressure, Pa.

Air's density largely depends on its temperature and pressure. The present analysis uses the atmospheric pressure, as the working pressure difference in the cooling unit is about 10–30 Pa, which is 0.01–0.03%. This analysis uses approximations of lower accuracy. Taking differential pressure of this order into account would complicate the analysis in the pursuit of imaginary accuracy. The air density of the boundary layer equals

$$\rho_l = \frac{p}{R(t_w + 273)}, \tag{9}$$

where $R$—gas constant, J/(kg·K).

V.P. Gavrikin and E.A. Kuranov obtained the formula to determine the wet-bulb temperature [35] resulting from the analysis of reference data [30]:

$$t_w = \frac{-7.14 + 0.651 \cdot l}{1 + 9.7 \cdot 10^{-3} \cdot l - 3.12 \cdot 10^{-6} \cdot I^2}, \tag{10}$$

where $I$—enthalpy (heat content) of the air, kJ/(kg·K).

The results obtained using the formula deviate from the tabulated values within 0.6%, which is quite sufficient for an engineering analysis.

Water-evaporative cooling takes place with the constant heat content, taking into account that enthalpy of humid air depends on the temperature and moisture content. Taking into account the availability of all initial data for outdoor air, it will be convenient to determine its heat content. The enthalpy is to be determined from the formula

$$I = c_{d.a.} \cdot t_o + \frac{(r + c_v \cdot t_o) d_o}{1000}, \tag{11}$$

where $c_{d.a.}$—specific mass heat capacity of dry air, kJ/(kg·K).

On the other hand, in a steady-state process the heat flux can be defined as the vaporization heat of the moisture evaporated in the cooling unit, which is expressed by the following formula:

$$q_{vap} = \frac{W \cdot \overline{\rho} (d_{su} - d_o)(r + c_v \cdot t_w)}{3600 \cdot 10^3}, \tag{12}$$

where $W$—cooling unit capacity, m$^3$/h; $\overline{\rho}$—average density of the cooled air, kg/m$^3$; $d_{su}$—moisture content of air at the cooling unit outlet, g/kg d.a.; $d_o$—moisture content of outdoor

air, g/kg d.a.; $c_v$—specific mass heat capacity of dry vapor, kJ/(kg·K) and $r$—specific heat of water vaporization at 0 °C kJ/kg.

Relationship (12) is of a deterministic nature, and its accuracy is absolute under conditions of initial data reliability. The specific humidity of the air at the unit inlet and outlet is measurable. However, in our case air parameters at the unit outlet are forecast. In a particular case, the expected moisture content of the air at the cooling unit outlet is equal to the one required by the technology. In all other cases, it is different, which is caused by the difference in the physical properties of the interacting media.

The surface may be covered with a continuous stable film (the film-wetting mode) or individual droplets or drips (the droplet-wetting mode), which determines the size of the interaction surface area. When hygroscopic materials are used, in addition to good wettability they have a certain buffer capacity, which can compensate for disturbances in the film mode.

The specific humidity of air is defined by the formula

$$d = 0.77 \frac{\varphi}{100} e^{5516.89\left(\frac{1}{273} - \frac{1}{273+t}\right)}, \tag{13}$$

where $\varphi$—relative humidity of the air, %; $e$—the basis of natural logarithm and $t$—the air temperature, °C.

A similar formula is used to determine the specific humidity of air at the cooling unit outlet.

The water-evaporative cooling process is iso-enthalpic. On L.K. Ramzin's I-d diagram, it appears as a segment connecting points 1 and 2, which characterize the state of moist air before (1) and after (2) entering the cooling unit (Figure 8). It is obvious that water-evaporative cooling is possible only if the following condition is observed: $d_1 < d_2$.

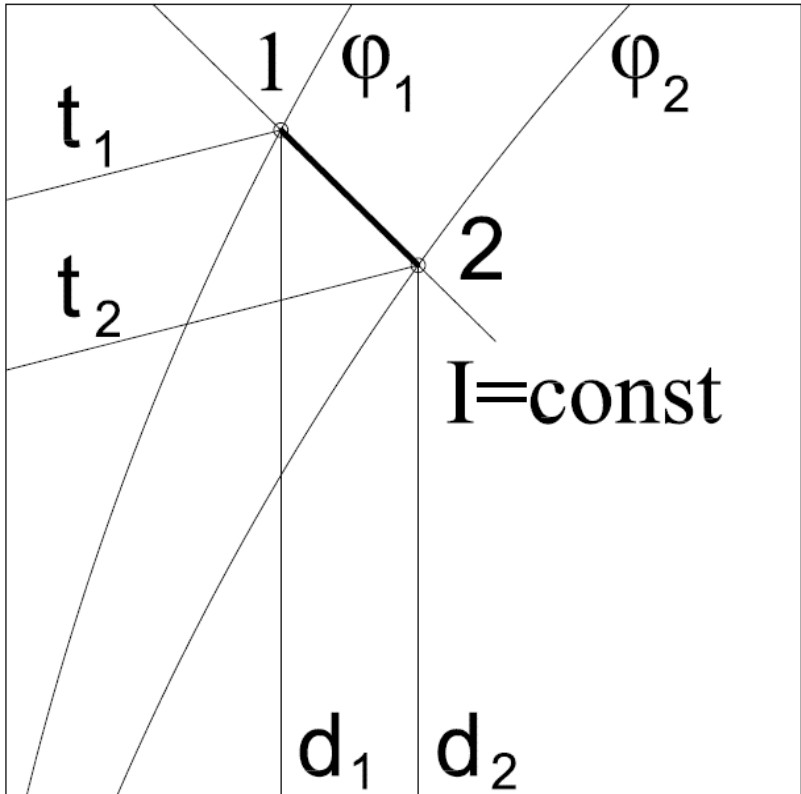

**Figure 8.** Water-evaporative cooling in the I-d diagram.

The average air velocity in the cooling unit is determined from the formula

$$\overline{v} = \frac{W}{3600 \cdot \frac{S}{2}},$$ (14)

where $W$—cooling unit capacity, m$^3$/h and $S$—cross-sectional area of the heat exchanger, m$^2$.

In the process of heat exchange, the air temperature decreases and its density increases proportionally. The average density of the cooled air can be determined from the following formula:

$$\overline{\rho} = \frac{p}{R\left(\frac{t_o + t_w}{2} + 273\right)}$$ (15)

The supply air temperature drop is proportional to the heat dissipated from it, provided that it does not exceed q$_v$ (the heat of vaporization, with a known difference in the specific humidity of the air) and can be found from the following system of conditions:

$$\begin{cases} \text{By} \, q_{\text{cool}} \leq q_{vap} \, t_{su} = \frac{q_{\text{cool}} \cdot 10^3}{0.28 \cdot W \cdot \overline{\rho}} \\ \text{If} \, q_{\text{cool}} \succ q_{vap} \, t_{su} = \frac{q_{vap} \cdot 10^3}{0.28 \cdot W \cdot \overline{\rho}} \end{cases}$$ (16)

By determining the air supply temperature and taking into account its specific humidity, we can obtain the relative humidity at the cooling unit outlet:

$$\varphi_{su} = \frac{100 \cdot d_{su}}{3.8 \cdot e^{5404.3 \cdot \left(\frac{1}{273} - \frac{1}{273+t}\right)}},$$ (17)

where $d_{su}$—moisture content of the air at the cooling unit outlet, g/kg of d.a.

To increase the practical value of the model, it is advisable to determine the minimum necessary water consumption for water-evaporative cooling with a given capacity under specific weather conditions, and the annual water consumption for water-evaporative cooling.

To do this, the wet flow rate for cooling one kilogram of air for specific weather conditions must be determined. Then, the sum of the required air exchange, the specific moisture consumption and the duration of specific weather conditions over a year will give the required value of the annual water consumption for water-evaporative cooling. Weather conditions vary from year to year, so climate data should be used in the design calculations.

The maximum hourly water consumption (m$^3$/h) for evaporation is determined from the formula

$$Q_w = W_{\text{max}} \cdot \overline{\rho} \cdot \Delta d_{\text{max}} \cdot 10^{-6},$$ (18)

where $W_{\text{max}}$—air exchange rate at maximum design outdoor temperature, m$^3$/h and $\Delta d_{\text{max}}$—the amount of moisture consumed by one kilogram of supply air during cooling at the maximum estimated outdoor temperature, g/kg of d.a.

Annual water consumption for water-evaporative cooling can be determined from the formula

$$Q_{w.a.} = \sum_{i=1}^{n} W_i \cdot \overline{\rho} \cdot \Delta d_i \cdot \tau_i \cdot 10^{-6},$$ (19)

where $W_i$—air exchange required at temperature $t_i$ (determined from the balance of humidity and heat), m$^3$/h; $\Delta d_i = d_{su} - d_o$—amount of moisture consumed by one kilogram of supply air during cooling, g/kg d.a. and $\tau_i$—duration of temperature period $t_i$, h.

For aggregate calculations, the values of relative humidity can be obtained by linear interpolation of reference points obtained from building regulations (BR) and local building norms (LBN). For more precise calculations it is necessary to use data tables (the relationship between temperature and relative humidity) or diagrams that take into account the duration of specific combinations of temperature and relative humidity for the considered climatic zone. Such data can be found in the works of A.Y. Kreslin and other scientists.

Knowing the required air exchange rate and the permissible total aerodynamic drag or limiting the flow speed with the condition that no droplets of moisture can escape from the sprayed surfaces, we can determine the required cross-sectional area, and, consequently, the required number of units.

### 2.3. Experimental Study

Description of the Experimental Facilities

The climate control system using the combined climate control unit was tested in fattening sector No. 6 on Farm No. 7 of "Firma Mortadel" LLC. Indicators of test conditions are given in Table 3.

**Table 3.** Parameters and conditions of the tests.

| Index | Value |
|---|---|
| Material of the building | Reinforced concrete |
| Holding capacity, heads | 250 |
| Internal dimension of the building L × W × H, m | 14.6 × 14.6 × 2.6 |
| Total floor area, m$^2$ | 214.3 |
| Area of the technological section, m$^2$ | 200.3 |
| Utilization factor of the floor area of the premises | 0.9 |
| Actual number of animals, heads | 279 |
| Floor area of the technological section per one animal, m$^2$/head | 0.7 |
| Characteristics of the livestock | |
| Age and sex group | Fattened pigs |
| Average weight, kg | 70 |
| Climatic conditions | |
| Region | Vladimir region |
| Estimated temperature of the coldest five-day period with a probability of 0.98, °C<br>Average monthly relative humidity at 3 p.m. of the warmest month, % temperature, °C | −34 |
| Average monthly relative humidity at 3 p.m. of the coldest month, % | 83 |
| Air temperature in summer, °C, probability 0.98 | 27 |
| Average monthly relative humidity at 3 p.m. of the warmest month, % | 57 |
| Indoor climate | |
| temperature, °C | 21 |
| relative humidity, % | 60 |

As a control sector, we used fattening sector No. 5, having 241 pigs with a weight of 30.5 kg (20 animals per pen). The climate maintenance system was designed by FG-CLIMITED, Ontario, CA, USA, using equipment from Ag-Co Products Limited Ontario, CA, USA. The front wall has four extractor fans (one extractor from the manure channel). The air is supplied through supply valves located in the ceiling and distributed in the room at the rate of one valve per box for a total of 12. The automation system is controlled by a temperature sensor installed in the room center at a height of 1.2 m above the floor level.

The experimental sector was additionally equipped with a tested system consisting of three prototypes of a combined climate control system with a water sprinkler cooling system (Figures 9 and 10): No. 1, 2—general exchange ventilation; and No. 3—air extraction from the manure channel and a system for regulating indoor climate parameters according

to temperature and relative humidity sensors placed along the passage at a height of 1.2 m. Technical characteristics of the units are presented in Table 4.

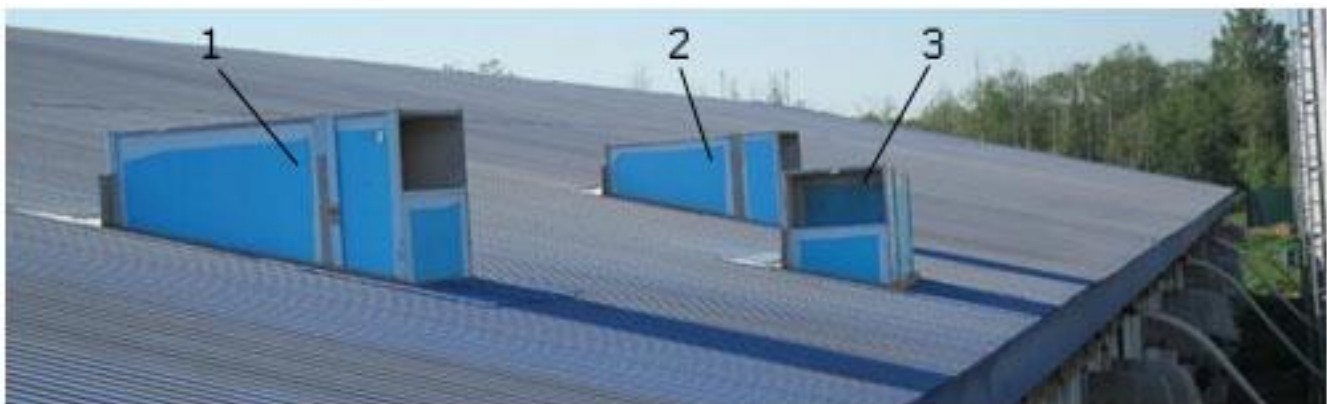

**Figure 9.** View from the pig house roof: 1,2—UT-6000S units (for general exchange ventilation); 3—UT-3000 unit (for air extraction from the manure channel).

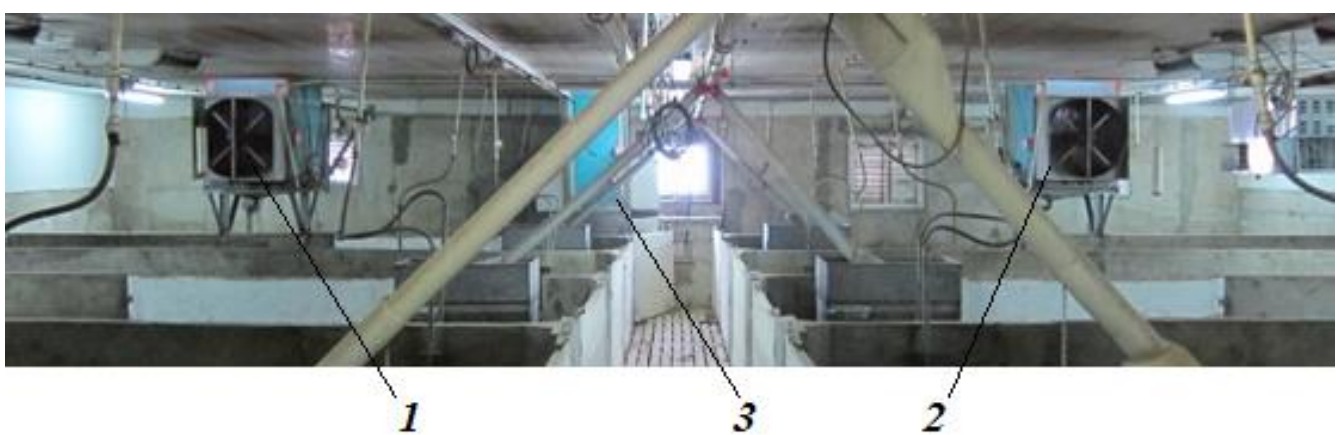

**Figure 10.** General view of the experimental system of climate control inside the pig house: 1,2—UT-6000C unit (for general exchange ventilation); 3—UT-3000 unit (for air extraction from the manure channel).

**Table 4.** Technical characteristics of the units.

| Designation | Value | |
| --- | --- | --- |
| Working designation | UT—6000C | UT—3000 |
| Air capacity, m³/h | 6000 | 3000 |
| Heat output at −25 °C, kW | 50.7 | 25.4 |
| Installed electric power, kW | 1.6 | 0.8 |
| Type | With defrosting system | Without defrosting system |
| Note | Reduced sump dimensions | With two supply fans |

The summer ventilation system was supplemented with an option to use the climate control units in the cooling mode.

During the test period, the outdoor air temperature ranged from −8 °C to +31 °C, and climate measurements were taken automatically by a climate control computer according to temperature and relative humidity sensors placed as shown in the diagram in Figure 11.

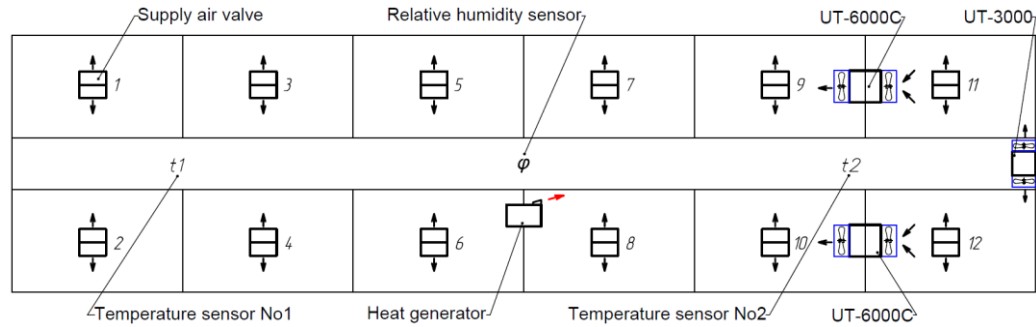

**Figure 11.** Control point diagram with sensors.

The climate control computer automatically maintains the specified indoor air temperature and relative humidity. Manual climate control is also possible.

As the relative humidity in the room increases, the air exchange rate smoothly increases as well. The air supply is controlled automatically by changing the speed of the fans and the degree of opening of the air inlet flaps. If the room temperature falls below the set point while the excess moisture is being removed, the heating devices are switched on. If the temperature drops further below the set point value, the ventilation level is set to the minimum value (depending on the weight and age of the animals), with the temperature taking priority.

This solution ensures the removal of excess moisture, heat and harmful gases from the house throughout the year.

## 3. Results

*Study of Water-Evaporative Cooling*

The problem was solved in Excel 2007 (Microsoft Corporation, Redmond, WA, USA).

Within the variation range of relative humidity from 20 to 100% and outdoor air temperature from 20 to 50 °C; the calculation results are graphically presented in Figure 12.

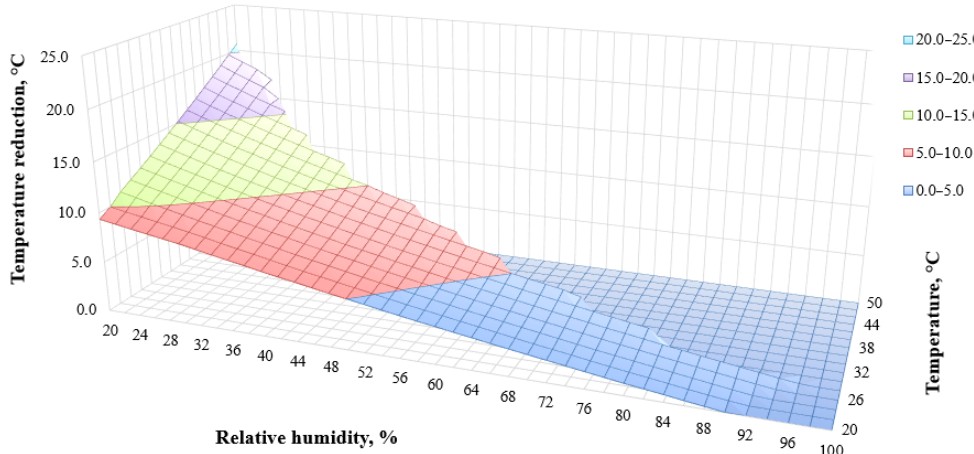

**Figure 12.** Reduction of air temperature in the unit as a function of the temperature and relative humidity of the outdoor air.

The graph is plotted in a three-dimensional coordinate system, showing the relationship of $\Delta t = f(t, \varphi)$. The result obtained shows that the maximum cooling value is achieved at minimum relative humidity and maximum outdoor air temperature, otherwise the supply air cooling value decreases monotonically.

Over the test period, the outdoor air temperature was 30.4 °C and relative humidity was 31.2%. The test results are shown in Table 5. Figure 13 presents a diagram of the exhaust and supply air temperatures.

**Table 5.** Test results of the combined climate control unit in the cooling mode.

| | Air Parameters in Control Points | | |
| Points | Indoor Temperature, °C | Difference between Indoor and Outdoor Temperatures, °C | Indoor Relative Humidity, % |
| --- | --- | --- | --- |
| 1 | 29.4 | 1.8 | 37.1 |
| 2 | 29.5 | 1.7 | 34.4 |
| 3 | 28.8 | 2.4 | 34.3 |
| 4 | 28.7 | 2.5 | 37.2 |
| 5 | 28.4 | 2.8 | 39.2 |
| 6 | 28.6 | 2.6 | 38.8 |
| 7 | 27.3 | 3.9 | 44.3 |
| 8 | 27.4 | 3.8 | 46.2 |
| 9 | 25.3 | 5.9 | 45.6 |
| 10 | 24.9 | 6.3 | 44.8 |
| 11 | 25.5 | 5.7 | 43.9 |
| 12 | 25.3 | 5.9 | 42.5 |
| Average: | 27.4 | 3. 8 | 40.7 |

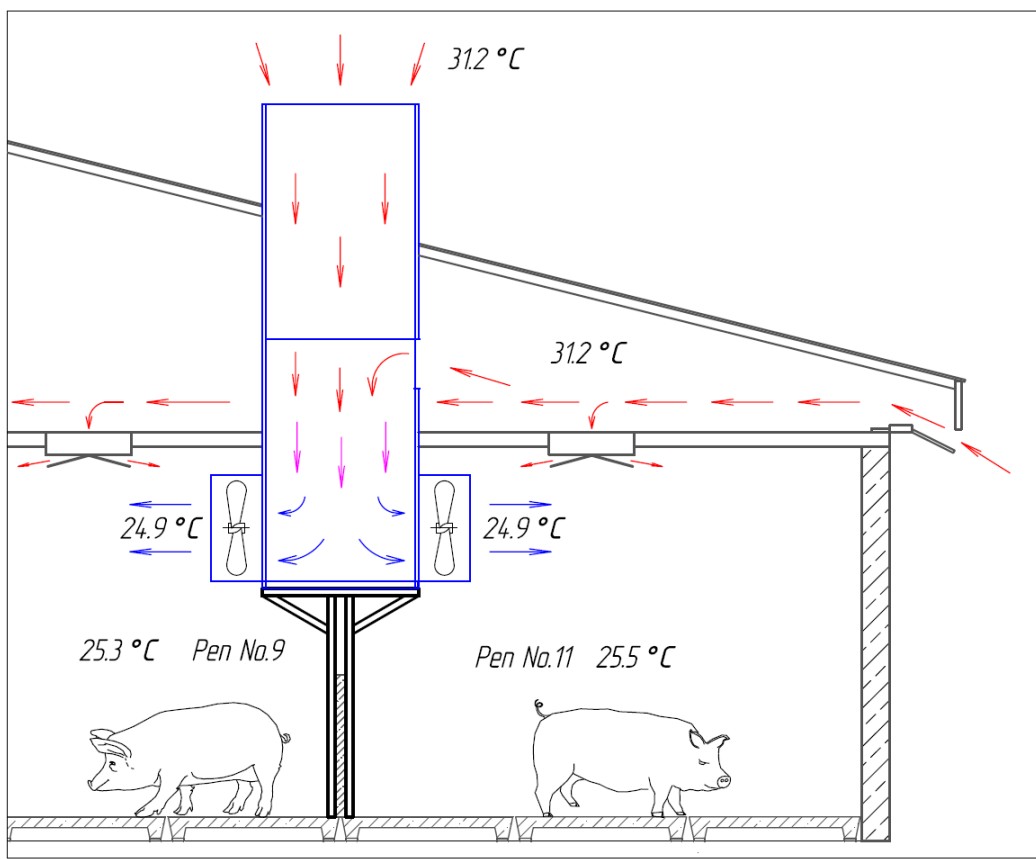

**Figure 13.** Diagram of the exhaust and air supply temperatures.

Table 5 shows that the maximum value of the air supply temperature reduction was 6.3 °C (in the pens closest to the units). As the distance from the units increased, the indoor air temperature increased. This effect was due to the fact that the share of cooled air (from

the units) in the total ventilation system was 49%. As a result, the cooled air mixed with untreated outdoor air from the general ventilation system. Therefore, the average room temperature drop was 3.8 °C.

When analyzing the convergence of the results, we noted some discrepancies with the analytical data. The calculation model shows that supplied air with a temperature of 31.2 °C and a relative humidity of 30.4% can be cooled by 8.3 °C when saturated with moisture to a relative humidity of 90.0% (by 11.7 °C at 100.0% relative humidity).

At small values of the coefficient of spraying heat-exchange plates, we registered reduced values of the air supply temperature as compared to the calculated ones. This effect is caused by the influence of the wettability of the heat exchange surface on the efficiency of water-evaporative cooling. When the water flow rate was increased, the experimental data came close to what was theoretically possible, but this was accompanied by a detachment of moisture droplets.

The stated assumption can be proven by the fact that in sprayed layers made of hygroscopic and well-wetted cellulose, the above effect was not observed in prcatice (under condition of a sufficient water flow rate for water evaporation).

Further research is required to study the influence of the wettability of sprayed surfaces on the efficiency of water-evaporative cooling.

## 4. Discussion

The problem of ensuring proper climate control in livestock buildings is rather urgent and requires special attention. A considerable amount of research focuses on the development and optimization of climate control systems to increase functionality and improve technological and energy efficiency. The authors aimed to develop and test an air cooling system in a combined climate control system used in pig production. To achieve this goal, we analyzed the existing cooling systems for ventilation air, developed the design of a climate control system and a mathematical model of air cooling, made a prototype and conducted experimental research.

In the course of our research, different methods of air cooling in climate control systems were considered. For livestock farming, it is advisable to use water-evaporative cooling systems with sprayed surfaces. This variant is optimal as these systems are characterized by constructive simplicity, reliability and technological and energy efficiency. As compared to other methods of air cooling, surface-filled water-evaporative cooling systems require a minimum amount of electricity per kW of cooling. This was the main argument for the use of the water-evaporative cooling system with sprayed surfaces in the combined climate control unit.

The authors developed a mathematical model describing air cooling for the offered technical solution. The model considers a heat exchanger consisting of tubes with a rectangular cross-section. The flow of the cooled air and the refrigerant water is direct in the considered heat exchanger. The water sprayed by the nozzles wets the surface of the heat exchanger. This makes it possible to obtain a large cooling surface, which has a positive effect on cooling intensity. Based on the obtained mathematical model, we determined the theoretical values of air temperature reduction depending on outdoor air temperature and humidity.

The developed combined climate control unit with a water-evaporative cooling system was tested on the "Firma Mortadel" Ltd. farm. In the course of testing, measurements of temperature and relative air humidity were taken in the animals' pen according to the conventional procedure, and the test results confirmed expectations. The developed system decreases the air temperature.

The prototypes were tested in the operating heating and ventilation system. The capacity of the three test units was 15,000 m$^3$/h, which provided 49% of the total capacity of the combined system. Since all the cooling units were concentrated on one side, the cooled air was unevenly distributed inside the space. To reduce this effect, it is advisable to distribute the units evenly over the space and cool the whole air supply.

The efficiency of water-evaporative cooling systems strongly depends on the climatic conditions in the region of operation. The highest efficiency is achieved in hot and arid as well as in temperate climates. Under conditions of high relative humidity—80% and above—the cooling effect tends to zero. As the cooling effect increases, so does water consumption. For example, at an outside air temperature of 40.0 °C and a relative humidity of 20.0%, it is possible to warm air to a temperature of 28.8 °C and ensure a relative humidity of 85.0%. Under these conditions, a unit with a capacity of 6000 m$^3$ of air per hour will have a water consumption of 23 m$^3$/h.

## 5. Conclusions

The analysis of the existing air cooling systems has shown that it is advisable to use water-evaporative cooling systems for livestock houses due to their simplicity, technological and energy efficiency and low requirements for operating conditions.

The comparative analysis of the energy efficiency of cooling systems has shown that systems with sprayed surfaces require a minimum amount of electric power to produce 1 kW of cold (0.003 kW/kW).

The combined climate control system includes a heat exchanger with a large heat exchange surface. Adding a heat exchanger spraying system is therefore sufficient to ensure water-evaporative cooling in the system. This system can also perform cleaning and disinfection of the unit's working surfaces as well as the inlet and outlet air of the serviced area.

To describe the cooling process in the combined climate control unit, the authors have developed a mathematical model that links air parameters at the inlet and outlet of the unit, its geometrical characteristics and performance.

The mathematical model can determine the potential cooling of the air supply and the minimum water consumption required to produce this effect. An air supply with a temperature of 31.2 °C and a relative humidity of 30.4% can be cooled by 8.3 °C when saturated with moisture to a relative humidity of 90.0% (by 11.7 °C at 100.0% humidity).

As the experimental research proved, the air supply was cooled by 6.3 °C (in the pens nearest to the units), which confirms the efficiency of water-evaporative cooling. However, additional research is required to study the influence of the wettability of the sprayed surfaces on the efficiency of water-evaporative cooling.

## 6. Patents

Heat recovery unit. Ilyin, I.V. and Ignatkin, I.Yu. Patent for the invention No. 2627199 C1 RU, MICROPK A01K 29/09 (2006/01). Applicant and patent holder—Russian State Agrarian University—Moscow Timiryazev Agricultural Academy—No. 2016127599; applied on 8 July 2016; issued on 3 August 2017, Bulletin No. 22-6 p.: ill.

**Author Contributions:** Software, D.S.; Validation, S.K.; Data curation, N.S. (Nikita Serov); Writing—original draft, I.I.; Writing—review & editing, N.S. (Nikolay Shevkun) and A.A.; Visualization, V.P.; Project administration, I.I. All authors have read and agreed to the published version of the manuscript.

**Funding:** This article was funded by an intra-university grant from the Russian State Agrarian University—Moscow Timiryazev Agricultural Academy "Food Sovereignty" section for research projects in the field of import substitution within the framework of the "Priority 2030" strategic academic leadership program.

**Institutional Review Board Statement:** Not applicable.

**Informed Consent Statement:** Not applicable.

**Data Availability Statement:** The data presented in this study are available on request from the authors.

**Conflicts of Interest:** The authors declare no conflict of interest.

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
