# Peer review of "Developing and Testing the Air Cooling System of a Combined Climate Control Unit Used in Pig Farming"

_agriculture, doi:10.3390/agriculture13020334_

Round 1

Reviewer 1 Report

General Comments

The manuscript is often incomprehensible and don’t meet any requirement of a paper in Agriculture (MDPI) and more generally of a scientific paper. If the model and experimental results seem valuable, it is fully devalued by its presentation, and it should be completely reorganized and rewritten.  I hope that the detailed comments below could help to undertake it.

Detailed Comments

L 53-60: Ultimately your objective is to provide better climatic conditions to pigs in farms. However, you never address this question with respect to pig’s needs in terms of climate, both for animal welfare, diseases, and parasites susceptibility, both inducing adequate conditions for pig growth, health and product quality. It deserves at least a mention and a short analysis in the introduction and to take it into account in the choice of your solutions.

L 61: You focus immediately to a single climate-parameter: the temperature, whereas air humidity, CO2, CH4 and pollutant contents can also interfere with pig’s climate needs.

L 63: Briefly explain the principle of vapor-compression refrigeration units (VCRU).

L 68: idem for water-evaporative cooling system.

L 79-85: You employ the notion of energy cost usually used for monetary purposes whereas you refer to a question about energy. Isn’t it an energy efficiency rather than  an energy cost? In addition, in table 1 you don’t use “energy cost” but “ energy efficiency”,  all that is not very clear, please clarify!

Table 1: Explain how are elaborated (calculated?) the elements of the table.

L 87 & Fig. 1: After “energy cost”, energy efficiency, you use “specific power consumption”, does they all correspond to the same thing?

Fig 1: is redundant with Table 1 and therefore useless! Simply comment line 9 of Table 1.

L 90: You mention “The calculations carried out” but you never present these calculations? See the previous remark on Table 1!

L 92: You mention “However, if strict temperature limits are set…,” You should before give a brief overview of pig’s needs in terms of climate (see remark/L 61) and particularly their tolerances.

L 97-99: At the very end of the introduction, you present the principle of VCRU whereas it should be presented at the beginning of your introduction (see remark L 63), as well as the other system.

L 38-99 & Table 1 & Fig. 1: The introduction must be rewritten, considering all the previous (and the next one) remarks.

L 100: Materials and Methods.

L 101-108: These considerations rather belong to the introduction.

Fig. 3: What is the meaning of “Spraing” ? Spraying?

L 129: Which order of magnitude, be precise!

Fig. 4 & corresponding text: It is not yet clear if the system described is a prototype developed by the authors or a commercial system? In addition, your figure shows 3 operating modes whereas you mention in the text, up to now, only one: the evaporative cooling. Consequently, remove or comment the 2 other ones. Also, your comments of the functioning have to be more detailed, the simple enumeration of the system’s components is not enough.

L 153- 163 & Fig. 5: There is no explanation of the figure, nor of the symbol used and meaning of the abscise and ordinate, please correct! What are t1, t2, tM, tau, tau, c ?

L 162-163: Explain how you can find a solution; it is the very objective of the Method section!

L 205: Cyrillic characters?

L 164-299: Most of the Result section is in fact a Method section?  You should reorganize completely your paper to be in line with the standards of a classical scientific paper! It is almost impossible to make an efficient review of this paper when all is mixed without any logic in the presentation!

Fig. 8 & 9 & 12/ These Figures are not quoted ore used in the text?

Table 2: Line in Cyrillic characters?

L 301-331:  The description of the experimental facilities that you put in the results should belong in fact to the method section, with 2 sub-sections: the experimental procedures and the theoretical or model approach (your previous section).

L 355: Line in Cyrillic characters?

L 349-385: The section “Study of water-evaporative cooling in the combined climate control unit in summertime” is in fact the results section and I state that there is no discussion.

L 354: Fig 11 is completely incomprehensible: No explanations of abscissa & ordinate & third dimension, only 1 color for different shades etc…

355: Cyrillic characters?

L 361: What is the meaning of pen?

L 366 and Table 6: Suppress the outside climate conditions from the table and put them in the Fig 6 captions. Also explain what t, phi and Delta t. Normally Figs and Tables must be autonomous and they should not necessitate to read the text.

L 373-376: You mention a comparison between model and experiment, but you never present it? Whereas you devote a lot of text to the presentation of the experimental devices?

L 387-416: The labeling 1, 2, 3, etc... in the conclusion is superfluous.

L 408-413: It should be part of the result or discussion (missing in your text) section rather than in the conclusion!

Author Response

Hello. I send the corrected article and answers to comments

Detailed Comments

L 53-60: Ultimately your objective is to provide better climatic conditions to pigs in farms. However, you never address this question with respect to pig’s needs in terms of climate, both for animal welfare, diseases, and parasites susceptibility, both inducing adequate conditions for pig growth, health and product quality. It deserves at least a mention and a short analysis in the introduction and to take it into account in the choice of your solutions.

Response to comment is provided in L 35-109 - we added material on the effect of microclimate on animal productivity. The tables and figures have been renumbered due to the material added.

L 61: You focus immediately to a single climate-parameter: the temperature, whereas air humidity, CO2, CH4 and pollutant contents can also interfere with pig’s climate needs.

Response to comment is given in L 41-109 - regulatory requirements for temperature, relative humidity, CO2, H2S for the Russian Federation and European countries have been added.

L 63: Briefly explain the principle of vapor-compression refrigeration units (VCRU).

Response to the comment is provided in L 143 - 152 – we added the working principle of steam compression refrigeration plants.

L 68: idem for water-evaporative cooling system.

Response to the comment is provided in L 153-156 – we added information on the principle of water evaporative cooling system.

L 79-85: You employ the notion of energy cost usually used for monetary purposes whereas you refer to a question about energy. Isn’t it an energy efficiency rather than  an energy cost? In addition, in table 1 you don’t use “energy cost” but “ energy efficiency”,  all that is not very clear, please clarify! Table 1: Explain how are elaborated (calculated?) the elements of the table.

Response to the comment is given in L171-173 - To estimate the energy efficiency of microclimate systems, we have chosen the indicator - the electrical power needed to produce 1 kW of cold. Additional figures from table 3 have been removed, Figure 2 duplicating Table 3 has been deleted.

L 87 & Fig. 1: After “energy cost”, energy efficiency, you use “specific power consumption”, does they all correspond to the same thing? Fig 1: is redundant with Table 1 and therefore useless! Simply comment line 9 of Table 1.

Response to the comment is given in L171-173 - To estimate the energy efficiency of microclimate systems, we have chosen the indicator - the electrical power needed to produce 1 kW of cold. Additional figures from table 3 have been removed, Figure 2 duplicating Table 3 has been deleted.

L 90: You mention “The calculations carried out” but you never present these calculations? See the previous remark on Table 1!

Table 3 is corrected to evaluate the energy efficiency of the climate control systems. We have analyzed  the electric output needed to produce 1 kW of cold, the additional figures in table 3 are deleted

L 92: You mention “However, if strict temperature limits are set…,” You should before give a brief overview of pig’s needs in terms of climate (see remark/L 61) and particularly their tolerances.

Response to the comment is given in L 41 - 109 - normative requirements for temperature, relative humidity, CO2, H2S in the Russian Federation and European countries were added.

L 97-99: At the very end of the introduction, you present the principle of VCRU whereas it should be presented at the beginning of your introduction (see remark L 63), as well as the other system.

Response to comment is given  in L 143-152 - material was replaced and supplemented.

L 38-99 & Table 1 & Fig. 1: The introduction must be rewritten, considering all the previous (and the next one) remarks.

Response to the comment is provided in L 25-201. Introduction section has been supplemented and corrected.

L 100: Materials and Methods. L 101-108: These considerations rather belong to the introduction.

Fragment moved to introduction L193-201.

Fig. 3: What is the meaning of “Spraing” ? Spraying?

This is a typing error, corrected.

L 129: Which order of magnitude, be precise!

Response to comment is provided in L 223-227. Clarified indicators are provided.

Fig. 4 & corresponding text: It is not yet clear if the system described is a prototype developed by the authors or a commercial system? In addition, your figure shows 3 operating modes whereas you mention in the text, up to now, only one: the evaporative cooling. Consequently, remove or comment the 2 other ones. Also, your comments of the functioning have to be more detailed, the simple enumeration of the system’s components is not enough.

Response to comment is given in L 228 – 235. We corrected Figure 4 (now Figure 6 in the text), added a description of the operating principle.

L 153- 163 & Fig. 5: There is no explanation of the figure, nor of the symbol used and meaning of the abscise and ordinate, please correct! What are t1, t2, tM, tau, tau, c ?

Response to the comment is given in L 259 – 261. Figure 5 has been modified (Figure 6 in the text), markings of values, axes of co-ordinates, decoding of symbols have been added

L 162-163: Explain how you can find a solution; it is the very objective of the Method section!

The phrase is deleted, the text is changed.

L 205: Cyrillic characters?

Response to comment is in L297-298 The text fragment has been translated.

L 164-299: Most of the Result section is in fact a Method section?  You should reorganize completely your paper to be in line with the standards of a classical scientific paper! It is almost impossible to make an efficient review of this paper when all is mixed without any logic in the presentation!

Response to comment is given in L242-458 text has been corrected and replaced to Materials and Methods,

Fig. 8, 9 & 12/ These Figures are not quoted ore used in the text?

Response to the comment is given in L414, 475 Reference in the text to Figures 8, 9, 12 were indicated and figures were renumbered (10, 11, 14).

Table 2: Line in Cyrillic characters?

Response to the comment is given in L403-404 – the text fragment has been translated.

L 301-331:  The description of the experimental facilities that you put in the results should belong in fact to the method section, with 2 sub-sections: the experimental procedures and the theoretical or model approach (your previous section).

Response to comment is given in L397-456 – the material has been replaced to "Methods" section. Two parts are highlighted in the section.

L 355: Line in Cyrillic characters?

Response to comment is given in L 469 - the text fragment has been translated.

L 349-385: The section “Study of water-evaporative cooling in the combined climate control unit in summertime” is in fact the results section and I state that there is no discussion.

Response to comment is given in L, 460-513, the material has been replaced to the Results section, the discussion section changed L 504 - 533.

L 354: Fig 11 is completely incomprehensible: No explanations of abscissa & ordinate & third dimension, only 1 color for different shades etc…

Response to comment is given in L466 – we added the designation of the abscissa and ordinate axes, the graph has been coloured

355: Cyrillic characters?

Response to the observation is provided in L469 Cyrillic text was deleted because it was duplicated

L 361: What is the meaning of pen?

We used “Pen” in the meaning of “a box for keeping pigs”.  Response is given in L 476 The scheme of air flows in Fig. 12 has been changed and the number of the figure has been changed to 14.

L 366 and Table 6: Suppress the outside climate conditions from the table and put them in the Fig 6 captions. Also explain what t, phi and Delta t. Normally Figs and Tables must be autonomous and they should not necessitate to read the text.

Response to comment is given  in L 280, 479-481.

L 373-376: You mention a comparison between model and experiment, but you never present it? Whereas you devote a lot of text to the presentation of the experimental devices?

Response to comment is given in L 588-592. Results of calculations of supply air temperature reduction as a function of outdoor air parameters are presented in Figure 13 (L 566)

L 387-416: The labeling 1, 2, 3, etc... in the conclusion is superfluous.

Response is provided in L 535-559, markings 1, 2, 3 are deleted.

L 408-413: It should be part of the result or discussion (missing in your text) section rather than in the conclusion!

Response to comment is provided in L 482-487, we have revised discussion section L504-533.

Best regards, 

the Author's.

Author Response

Hello. I send the corrected manuscript.

  1. The introduction section is too small and needs to be added.

The introduction has been expanded.

  1. Fig 11 has no units and the presentation is confusing.

Units added.

  1. Parts of it are in Russian (Eg. 205) and require the attention of the author.

Removed.

  1. Supplementary test pig house air supply and exhaust system diagram.

The same, but without prototypes.

  1. The results and discussion are too little, and the comparison between simulated and measured values needs to be presented.

Results and discussion expanded, comparison added.

  1. Where is the winter heat exchanger reflected.

In this publication, we focused on testing a water-evaporative cooling system and corrected the name.

  1. It is recommended to supplement the outdoor temperature curve with a small amount of data from indoor temperature measurement points.

Corrected figure 8 and table 6.

Best regards, 

the Author's.

Reviewer 3 Report

The paper titled ‘Developing and testing a combined climate control unit with heat recovery and the air cooling system for pig farming’. The main objectives of this research are to develop and test a combined climate control system with heat recovery and air cooling to be used in fattening pigsties. The experimental apparatus is described and the adopted method is discussed. However, the description of the system development process is not very detailed. Specific Comments 1. The innovation and contribution of this article should be noted in the abstract, and the abstract should be as concise as possible 2. Limit the use of lumped references ([1-3], [4-8],[9-24]), but explain each reference contribution. 3. The introduction and development of such systems should be added in the introduction section. DOI: 10.1016/j.renene.2021.06.082 DOI: 10.1016/j.enconman.2022.115579 4. Do ejector and nozzles have electric power consumption(Fig. 2)? They have no power input, so I don't think they will consume power. 5. Please provide the specific parameters of the system. 6. Please add the subtitle to Fig. 4. 7. The equations should be presented, in a numerical section, before the results. 8. Conclusions. More quantitative conclusions should be presented.

Author Response

Hello. I send responses to comments and a revised manuscript.

The paper titled ‘Developing and testing a combined climate control unit with heat recovery and the air cooling system for pig farming’. The main objectives of this research are to develop and test a combined climate control system with heat recovery and air cooling to be used in fattening pigsties. The experimental apparatus is described and the adopted method is discussed. However, the description of the system development process is not very detailed. Specific

Response to the comment is provided in L 14-21, L 228-235 – we have described the plant operation in the cooling mode.

Comments 

  1. The innovation and contribution of this article should be noted in the abstract, and the abstract should be as concise as possible 

Response to comment is provided in L 14-21, we have revised and shortened abstract

  1. Limit the use of lumped references ï¼ˆ[1-3], [4-8],[9-24]), but explain each reference contribution. 

Response to comment: references have been regrouped at the request of the reviewer.

  1. The introduction and development of such systems should be added in the introduction section. 

DOI: 10.1016/j.renene.2021.06.082 

DOI: 10.1016/j.enconman.2022.115579 

Response to the comment is given in L120-121 – We have added references to articles recommended by the reviewer (16 and 17 literature sources).

  1. Do ejector and nozzles have electric power consumption(Fig. 2)? They have no power input, so I don't think they will consume power. 

You are right, the ejectors do not consume electrical power directly. However, in order to make them work, water must be supplied at a given flow rate and pressure, and this requires some mechanical work. Performing this work requires an expenditure of electrical energy at the pumping station. These figures have been taken into account when calculating the electrical power consumption to produce 1 kW of cooling.

  1. Please provide the specific parameters of the system. 

The answer to the observation is given in L403, 433 system characteristics are given in Tables 5 and 6

  1. Please add the subtitle to Fig. 4. 

Response to the observation is presented in L237. We have edited the figure according to the operating principle.

  1. The equations should be presented, in a numerical section, before the results. 

Response to this observation is given in L 203-395 the equations have been moved to "Methods"

  1. Conclusions. More quantitative conclusions should be presented.

Response to the comment is given in L 535-560 - quantitative conclusions of the work are presented.

Best regards, 

the Author's.

Round 2

Reviewer 1 Report

Indeed, there is a progress with respect to the first version of the manuscript! However, the substance of the paper is too frequently hidden by a form which is still deficient! Tenth of words are glued each other making impossible the understanding of the text! Several tables and figures are either useless or incomprehensible. The job of a reviewer is not to correct the spelling of the papers but to judge about its substance. Consequently, I invite the authors to be next time irreproachable in this field, otherwise I will ask for a reject. Attached you will find the manuscript with some corrections.-

Author Response

Dear reviewer, thank you very much for your comments!

We have done our best to make all possible corrections and smooth out the text.

In particular:

  • We have left one digit after the decimal point in temperature values (we used to have one or two)
  • Figure 2 and lines 53-55 have been removed.
  • Figure 3 – we have added the pig breed - the Large White
  • We have also explained in plain language what RD_APK and DIN 18910 are – these are normative documents regulating the indoor climate requirements for pig farms and facilities in Russia and Germany, respectively.
  • Tab 1 - The optimum values for temperature and relative humidity were taken from normative documents (RD-APC 1.10.02.04-12 and DIN 18910)
  • 2 has been deleted.
  • Comment on heading 2.1 - Theoretical research - All text below refers to the description of the system, but not to the theoretical research : subsection 2.1 "System description" has been introduced, followed by subsection 2.2 “Theoretical research" with the formulae, and subsection 2.3 “Experimental research".
  • The designation of the value “de.p” has been unified in formulae 2, 3, 5.
  • Reference source 32 (pasted into the text) has been added to formula (3).
  • Table 6, last column – we have clarified that we mean the relative indoor humidity there!
  • Discussion – we have clarified the characteristics and described the existing limitations of the system being tested.
  • In table 4 (previously 5), we have replaced the Russian letters with UT.
  • We have also replaced Figure 11 (previously 12).
  • In lines 89-91 we have added the explanation for the carbon dioxide concentration, the table has been removed and it is now unclear why it is referring to concentrations of 0.2 and 0.3 %.
  • We have also split all the glued words, unified the terms, checked the spelling and punctuation.

We do hope that our modified version is clearer and more logical now!

P.S.

I send the manuscript in PDF format.

MS Word version sent by Yvan Chen Assistant Editor.
